# Internet Use Impact on Physical Health during COVID-19 Lockdown in Bangladesh: A Web-Based Cross-Sectional Study

**DOI:** 10.3390/ijerph182010728

**Published:** 2021-10-13

**Authors:** Tanvir Abir, Uchechukwu Levi Osuagwu, Dewan Muhammad Nur-A Yazdani, Abdullah Al Mamun, Kaniz Kakon, Anas A. Salamah, Noor Raihani Zainol, Mansura Khanam, Kingsley Emwinyore Agho

**Affiliations:** 1School of Business, Ahsanullah University of Science and Technology, Dhaka 1208, Bangladesh; 2Translational Health Research Institute (THRI), School of Medicine, Western Sydney University, Campbelltown, NSW 2560, Australia; l.osuagwu@westernsydney.edu.au (U.L.O.); K.Agho@westernsydney.edu.au (K.E.A.); 3African Vision Research Institute (AVRI), Westville Campus, University of KwaZulu-Natal, Durban 3629, South Africa; 4Research Associate, Creative Research & Consultancy, Dhaka 1205, Bangladesh; dewanyazdani@gmail.com; 5Faculty of Business and Management, UCSI University, Kuala Lumpur 56000, Malaysia; abdullaham@ucsiuniversity.edu.my; 6Department of Philosophy, College of Arts and Sciences—CAAS, International University of Business Agriculture and Technology—IUBAT University, Dhaka 1230, Bangladesh; kanizk2@yahoo.com; 7Department of Management Information Systems, College of Business Administration, Prince Sattam Bin Abdulaziz University, 165, Al-Kharj 11942, Saudi Arabia; a.salameh@psau.edu.sa; 8Faculty of Entrepreneurship and Business, Universiti Malaysia Kelantan, Kota Bharu 16100, Malaysia; raihani@umk.edu.my; 9International Centre for Diarrhoeal Disease Research, Bangladesh (Icddrb), GPO BOX 128, 68, Shaheed Tajuddin Ahmed Sarani, Dhaka 1000, Bangladesh; mansura@icddrb.org; 10School of Health Sciences, Western Sydney University, Campbelltown, NSW 2560, Australia

**Keywords:** internet, coronavirus, lockdown, headache, back pain, neck pain, physical health

## Abstract

Previous studies on internet use frequency were focused on mental health impact, with little known about the impact on physical health during the COVID-19 lockdown. This study examined the impact of internet use frequency on self-reported physical health during the COVID-19 lockdown in Bangladesh. A web-based cross-sectional study on 3242 individuals was conducted from 2 August–1 October 2020. The survey covered demographics, internet use frequency and self-reported physical health questions. Linear regression analyses were used to examine the impact of internet use frequency on physical health. 72.5%, 69.9%, 65.1% and 55.3% respondents reported headache, back pain, numbness of the fingers and neck pain, respectively. The analyses showed increased physical health impact among regular (coefficient β = 0.52, 95% confidence interval [CI]: 0.18–0.85, *p* = 0.003), frequent (β = 1.21, 95% CI: 0.88–1.54, *p* < 0.001) and intense (β = 2.24, 95% CI: 1.91–2.57, *p* < 0.001) internet users. Other important predictors were gender, income, occupation, regions, and working status. Frequent and extensive uses of the internet were strong predictors of physical health problems, and our findings suggest the need for increased awareness about the physical health problems that can be triggered by excessive internet usage.

## 1. Introduction

The novel coronavirus disease 2019 (COVID-19) lockdown and the associated public health interventions, including social distancing, home quarantine measures, and ‘stay-at-home’ orders put in place by the various governments led to a surge in internet usage during the lockdown [1]. The lockdown measures resulted in a widespread unprecedented social disruption [2] as non-essential businesses were closed, some employees worked from home and unemployment increased [3]. There were massive changes in the pattern of use and behavior of internet users and employees had to quickly adjust to the new “normal”—with meetings going completely online, office work shifting to the home, leading to new emerging patterns of work. These changes were sudden to most organizations, businesses, societies, or governments, leaving barely any time for organizations and people to plan for, prepare and implement new setups and arrangements. Society has had to adjust, try, experiment, and find ways that did not exist before the lockdown [1], and these may have implications on the physical health of individuals.

People are now spending even more time with technology while consuming news media, watching television, using social media to connect with others, utilizing lifestyle apps to shop for groceries and other consumer goods, and engaging in home workouts [2]. In Bangladesh, this kept parts of the economy going [4] as the country witnessed a boom in internet usage due to the fast-growing mobile internet and the government’s push for digitalization. Engaging in social media platforms is now a routine activity for children and adults, enhancing communication, social connection, and even technical skills [5]. Platforms such as Facebook and Twitter offer multiple daily opportunities for connecting with friends, classmates, and people with shared interests. The number of people using such sites has also increased dramatically, as shown in a recent poll [6]. A past study reported that about 22% of teenagers log on to their favorite social media site more than ten times a day, and more than half of adolescents log on to a social media site more than once a day [6,7]. Seventy-five percent of teenagers now own cell phones, 25% use them for social media, 54% use them for texting, and 24% use them for instant messaging [7].

According to the Bangladesh Telecommunication Regulatory Commission, there were 99,428 million (approx. 60.9% of the population) internet users in February 2020 [8]. Though there is anecdotal early evidence that this may lead to increased productivity, it has also led to increased stress [9,10] where employees must learn new technologies, be available for work at almost all times, stay with digital devices all the time, and cope with multi-tasking [1]. The use of video-conferencing and content delivery services such as Zoom and Akamai also increased [1]. Those working from home using this technology have found themselves under intense scrutiny and tension [11].

That a large part of the peoples’ social and emotional development is occurring while on the internet and cell phones may affect the physical health of individuals considering the long hours spent on the internet during the mandatory stay-at-home orders, which exposes the population to lots of misinformation [12] known to affect physical and mental health [13]. With such high use of information systems becoming the new normal in our society, this study was designed to examine the impact of internet usage frequency during COVID-19 lockdown on self-reported physical and mental health of Bangladeshi residents’. The study also determined the level of addiction to social media platforms used by the locals.

## 2. Materials and Methods

### 2.1. Study Population and Sampling

By convenience sampling, 3242 questionnaires were collected from adults aged 18 and over living in eight divisions across Bangladesh during the lockdown. All questionnaires were web-based, participation was voluntary, and each participant was informed about the study’s background through an online preamble. No reward or incentive was offered for participation. In order to ensure the validity of data to some extent, restricting the same IP address and device are applied to the system, which means the participant of the same IP address or device only could participate once.

### 2.2. Data Collection 

Data were collected using self-administered web-based questionnaires distributed to various groups of people via the E-link using numerous mailing lists and social network sites including Facebook, WhatsApp and Twitter. Due to the mandatory lockdown, no paper questionnaires were distributed, making it difficult to reach the very remote regions.

### 2.3. Demographic Variables

Demographic variables (Table 1 for an overview and details) used for the study analyses were based on previous studies [14,15,16]. Demographic variables included demographic characteristics (i.e., gender, age [categorized as 18–27, 28–37, 38 years and over], marital status [single, married, divorced/widowed], mothers level of education [higher education, bachelor, intermediate], place of residence, employment status [employed, unemployed/student], occupation [healthcare and non-healthcare worker], and income in Taka [lower, middle and high-income earners]. For public health purposes and for the everyday descriptive purposes, continuous variables such as age were categorized [17].

### 2.4. Dependent Variable

Ten items of physical complaints reported in Table 2 below were used to determine the impact of individual self-reported physical health on internet use. The respondents used a close-end response [‘Yes’ or ‘No’] to indicate whether they suffered from the following ten problems (back and neck pain, numbness in fingers, headaches, inability to sleep, dry eyes or other vision problems, poor nutrition, poor personal hygiene, weight gain/loss and loss of appetite] during or after pro-longed internet use. Binary scores for individual questions were summed to give a physical health score, which ranged from 0–10, and the Kuder–Richardson coefficient measuring internal consistency among the physical health scores ranged from 0.65 to 0.78, indicating a satisfactory level of reliability.

### 2.5. Main Study Factor

The main study variable was the respondents’ internet usage, including the average daily amount of time spent on the internet and dependence on networks. The frequency of internet use was categorized as ‘seldom’ (<1 h per day), ‘casual’ (≥1 and <3 h per day), ‘regular’ (≥3 and <5 h per day), ‘frequent’ (≥5 and <7 h per day), ‘intense’ (≥7 h per day) [14].

### 2.6. Statistical Analysis

All statistical analyses were performed in Stata version 14.1 (Stata Corp. 2015, College Station USA) mainly using descriptive statistics; the *t*-test used compares the means of two independent groups, one-way analysis of variance (one-way ANOVA) used compares the means of more than two independent groups and simple and multiple linear regression analysis was conducted to estimate the unadjusted coefficient and adjusted coefficient.

In the simple linear regression analysis, all confounding variables with a *p* < 0.20 were retained and were used to build a multiple linear regression model [18]. A manual elimination procedure was applied for multiple linear regression analyses to remove non-significant variables (*p* > 0.05). To examine internet use impact, the main study factor variable (Internet usage frequency) was added to all significant confounding variables after elimination processes. The main study factor and demographic variables associated with physical health scores (*p* < 0.05) were reported. In the regression analysis, we checked for homogeneity of variance and multicollinearity, including Variance Inflation Factors (VIF) and the VIF < 4 was considered suitable [19].

## 3. Results

### 3.1. Demographic Characteristics According to the Frequency of Internet Use among Adults in Bangladesh

In Table 1; the main demographic characteristics, including gender, education and age group, on the five groups extracted are shown. It is worth noting that the education in this paper refers to that of the mothers’ maximum educational level. The majority of participants were males (1976, 61.1%), married (2632, 81.2%) and employed (2864, 88.3%) in non-healthcare sectors (2567, 79.2%), and for many, their mothers had completed a bachelor’s degree or higher. About half of the respondents earned more than 70,000 Taka at the time of data collection (Table 1).

About two-thirds of the respondents that reported intense use of the internet were males, while females were more likely to report regular and casual use of the internet. Ages ranged from 17 to 65 years of age, with a mean of 32.9 years (SD = 6.8). More than half of intense and frequent internet users were aged 28–37 years, and a similar proportion of casual internet users were younger (18–27 years). Nearly all intense internet users were employed and working in the non-healthcare sector during the study.

Table 2 shows the prevalence and 95% CIs of the adverse effects on self-reported physical health complaints examined in this study in males and females. Headache 72.50% [95% CI 70.93%, 74.01%] was the prevalent complaint of the participants, while 69.87% [68.27%, 71.43%] and 65.10% [63.42%, 66.71%] complained of back pain and numbness of the fingers, respectively. Males reported a higher prevalence of back pain, numbness of the finger, headaches, and neck pains than females as the confidence intervals do not overlap (Table 2). By contrast, females reported a higher prevalence of poor nutrition, poor personal hygiene, and loss of appetite compared with men (Table 2).

Figure 1 shows the mean scores of physical health by internet use frequency in males and females. From the figure, the mean scores of physical health by intensive internet use were 5.6 for males and 5.4 for females. This physical health mean scores could be translated as percentage mean scores of males (56%) and females (54%), respectively. Physical health mean scores were lower among the casual user of internet (3.2 for females and 3.0 for males) than seldom users of internet (3.1 for females and 3.4 for males).

### 3.2. Unadjusted Analysis for the Association between Prolonged Internet Use and Self-Reported Physical Health Complaints

Table 3 shows the mean scores of physical health complaints and the unadjusted coefficients for the associations with the sociodemographic variables. From the table, the mean scores for physical health complaints varied across variables. The unadjusted coefficients revealed significantly higher physical health problems due to prolonged internet use in those aged >28 years, married, completed bachelor education, middle-high income earners and non-healthcare workers compared with other groups. In contrast, females, those who were divorced or widowed, those with intermediate education, non-workers/students and respondents who resided in Chittagong, Dhaka and Khulna during the study period had significantly lower mean scores of physical health complaints from prolonged internet usage compared with the other groups.

## 4. The Impact of the Frequency of Internet Use on Physical Health Scores

Figure 2 presents the unadjusted and adjusted odds ratio and their confidence intervals for the association between frequency of internet use and self-reported physical health symptoms during COVID-19 lockdown. In the unadjusted analysis, frequent and intensive internet users during COVID-19 lockdown significantly reported higher physical health problems than seldom users of the internet. After adjusting for the demographic variables (see Table 2 for an overview and details) used in this study, regular, frequent, and intensive internet users during COVID-19 lockdown reported significantly higher physical health problems than those who seldom used the internet (Figure 2).

The results of multiple regression analyses predicting the level of physical complaints among internet users under COVID-19 lockdown is presented in Table 4. Marital status, age, gender, place of residence, level of education, working status, occupation, and income level were all associated with the self-reported physical health complaints among internet users in Bangladesh (Table 4).

## 5. Discussion

A considerable body of literature has emerged over the past two decades assessing the relationship between problematic, excessive, and addictive use of the internet and individuals’ psychological and/or mental well-being. Comparatively, very little research has evaluated the relationship between various internet use frequencies and the individual’s physical health. Considering the surge in the use of the internet during the COVID-19 lockdown [1], this cross-sectional study investigated the relationship between the frequency of internet use and the self-reported physical health of Bangladeshi respondents. The study found that more than one in every two persons surveyed, reported that they used the internet for seven or more hours each day, which was associated with a 2.2 folds increase in the self-reported physical health problems compared with seldom users (<1 h per day). Headache, back pain and numbness of the fingers (reported by over seventy percent of internet users) were the predominant complaints reported by internet users. After adjusting for the frequency of internet use, we found that middle to high-income earners and those working in non-healthcare sectors experienced more physical health problems due to internet use. In comparison, students and widowed or divorced persons reported fewer physical health problems compared with other respondents.

This study found that headache was the most prevalent symptom reported by internet users in Bangladesh, followed by back pain and numbness of the fingers. These complaints were significantly increased among frequent and intense internet users, which is in line with the findings of significant association between extensive internet use and physical health problems in previous studies [20,21,22,23]. In Spain, headache was also the most common complaint reported by internet users during COVID-19 lockdown [23]. Cao et al. (2009), found that among adolescents, those who used the internet excessively were more likely to report psychosomatic symptoms such as lack of physical energy, physiological dysfunction, weakened immunity, emotional symptoms, behavioral symptoms and social adaptation problems and had lower life satisfaction compared with normal internet users [21].

Among school children and other graduate students in China [24] and Italy [25] experienced, studies have shown that excessive internet use was associated with increased psychological distress, during COVID-19 lockdown. Such addiction to the internet can result in the problem of self-care, difficulty in performing daily routine, and mental health effects of anxiety and depression [26]. The negative effects of excessive internet use on health may be related to the alterations in the individuals neuroanatomical and neurochemical mechanisms, including cortical thinning of various components of the brain and dopaminergic reward circuitry, during excessive use of the internet [26]. While it is imperative that such behavior, especially during a lockdown, remain at a moderate and regulated level, identifying the population at risk of the health consequences of problematic use of the internet is necessary for prevention [27], early intervention and improved work or school functions [28].

Further more, this study that a higher proportion of men than women used the internet excessively during the lockdown and thus experienced higher physical health problems, which was significant only when it interacted with the frequency of internet use. Previous studies among adolescents in Taiwan [29] and mainland China [30] found that male gender was a potential risk factor for excessive internet use. The association of gender and self-reported physical health complaints found in the present study could be explained by the patriarchal nature of the Bangladesh society, where women and girls rarely go outside of their homes even before the lockdown [31] but preferred to stay home and spend more time on the internet. Whereas these women may have developed adaptive strategies, it was a different experience for the male counterparts, who were used to staying outside while working prior to the lockdown. For these men, the lockdown measures and transition to work online and/or staying at home required them to make sudden adjustments and find ways to adapt which did not exist previously [1] as educational institutions, businesses and workplaces were shut down for such a long duration [32]. These adjustments may have had greater impact on their physical health.

Another finding of this study was the significant relationship between income class and occupational status (non-healthcare worker) with the physical health problems experienced by internet users. Earning above 30,000 Taka was associated with higher physical health problems due to internet usage after adjusting for the frequency of internet use. In line with this finding, Islam et al. [33] on the correlation between problematic (excessive) internet use (PIU) and lifestyle during COVID-19 lockdown using data from a large sample of Bangladeshi youth and adults (*n* = 13,525) found that younger age, higher education, cigarette smoking, more sleep, less physical activity, gaming and social media use were associated with PIU [33]. This research has showed the physical health concerns related to prolonged internet usage during COVID-19 lockdown. A similar association was found in this research between educational status, age and self-reported physical health complaints among internet users in Bangladesh; however, these associations were dependent on the frequency of internet use, which was not assessed in the previous study [33].

The finding that non-healthcare workers had higher physical health problems compared with healthcare workers in this study is consistent with the finding of poorer psychological well-being among non-health-care workers who used the internet to obtain information on COVID-19 compared with those who received COVID-19 information from medical staff in healthcare settings [34]. Although this research was unable to identify whether the non-healthcare workers in our study used the internet for COVID-19 information retrieval, it has been shown that the influx of misinformation around COVID-19 via online platforms had adverse effects on people’s health [34]. Such problematic internet use (PIU) can lead to clinical impairment, distress [35] and/or deterioration in financial, familial, social, educational and/or occupational domains [36].

This study has some limitations. First, the cross-sectional design of this study does not allow for causality or the direction of relationships to be determined. However, appropriate analysis of cross-sectional data represents a useful initial step in identifying associations between extensive internet usage and the physical health condition of the participants. Second, this study is also limited by the fact that using a physical health score of 0–10 points may violate some linear regression assumptions [37]. Third, as the responses were self-reported, it was not possible to verify the participants’ physical health complaints, which may lead to a response bias. Multiple assessments, interviews, and informants may have provided a richer and more thorough understanding of this topic. Fourth, since the survey was administered during the COVID-19 lockdown using an online platform, it is possible that some respondents, especially those living in rural areas with limited internet access or people addicted to other online platforms such as gaming or social media, may not have participated in this survey. Although these findings may not represent the opinion of all internet users, a face-to-face survey was not possible at the time due to the lockdown measures. Fifth, as with online surveys that non-probability sampling, the statistical inference is difficult to do and the respondents are more likely to be those who have strong opinions about the subject. Despite these limitations, this was the first study to investigate the associations between extensive internet use and physical health symptoms in a sample of adults in mainland Bangladesh. Moreover, the use of online surveys and the sampling technique was valuable for reaching the hard-to-reach (although internet connected) populations future studies could address nesting follow-ups in existing random samples that could provide timely access to longitudinal population-representative data, including non-numerical data.

## 6. Conclusions

In summary, the study found that uncontrolled use of the internet is prevalent among the respondents in Bangladesh as more than two-third reported excessive use of the internet during the lockdown, and this doubled the adverse effects on their physical health compared with seldom users. Headaches, back pain, numbness of the fingers and neck pain were predominant among internet users as spending 3 h or more each day on the internet led to a significant increase in the level of adverse physical health problems. The positive associations between physical health complaints and occupation (non-healthcare workers) and income class (middle to high-income earners), indicate the target group for interventions to minimize this growing epidemic. The health department of the Bangladeshi government has now recognized extensive internet usage as a serious public health problem. Arguably, it is time for the World Health Organization and health departments worldwide to enhance effective health policies to increase public awareness of extensive internet use, especially post-lockdown. This has the potential to affect millions of children and adults and, eventually, the societies and economies.

## Figures and Tables

**Figure 1 ijerph-18-10728-f001:**
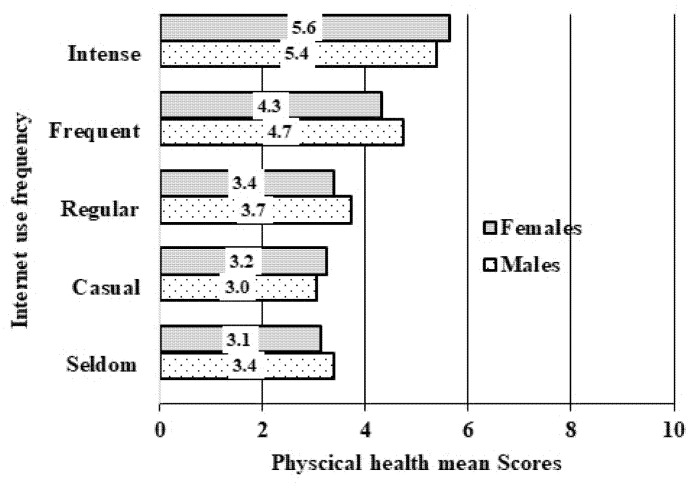
Mean scores of physical health under COVID-19 lockdown in Bangladesh, broken down by internet use and gender.

**Figure 2 ijerph-18-10728-f002:**
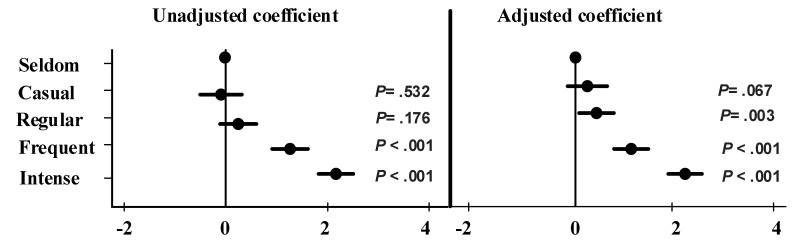
The impact of frequency of internet use on physical health scores: Unadjusted and adjusted coefficient and their 95% confidence intervals. Variables adjusted for were: gender, age group, division of residence, mother’s education, marital status, income, working status, occupation.

**Table 1 ijerph-18-10728-t001:** Demographic characteristics according to the frequency of internet use among adults in Bangladesh. Values are expressed as numbers and percentages (*n*, %).

Variables	Total (N = 3236)	Intense (*n* = 1668)	Frequent (*n* = 689)	Regular (*n* = 559)	Casual (*n* = 187)	Seldom (*n* = 93)
**Demography**						
*Gender*						
Male	1976 (61.1)	1179 (70.7)	382 (55.4)	264 (44.1)	93 (44.7)	58 (62.4)
Female	1260 (38.9)	489 (29.3)	307 (44.6)	335 (55.9)	94 (50.3)	35 (37.6)
*Age in years*						
18–27	555 (17.2)	30 (1.8)	156 (22.6)	254 (42.1)	98 (52.4)	17 (17.9)
28–37	1657 (51.1)	956 (57.3)	366 (53.1)	238 (39.5)	60 (32.1)	37 (39.0)
38+	1030 (31.8)	682 (40.9)	167 (24.2)	111 (18.4)	29 (15.5)	41 (43.2)
*Marital status*						
Single	508 (15.7)	29 (1.7)	150 (21.8)	223(37.0)	90 (48.1)	16 (16.8)
Married	2632 (81.2)	1632 (97.8)	505 (73.3)	339(56.2)	83 (44.40)	73 (76.8)
Divorced/widow	102 (3.1)	7 (0.4)	34 (4.9)	41 (6.8)	14 (7.49)	6 (6.3)
*Place of residence* (%)						
Barisal Division	172 (5.31)	70 (4.20)	48 (7.0)	35 (5.0)	7 (3.7)	12 (12.6)
Chittagong Division	291 (9.0)	100 (6)	107 (15.5)	61 (10.1)	16 (8.6)	7 (7.4)
Dhaka Division	1574 (48.6)	1010 (60.6)	205 (29.8)	236 (39.1)	95 (50.8)	28 (29.5)
Khulna Division	352 (10.9)	115 (6.9)	109 (15.8)	92 (15.3)	20 (10.7)	16 (16.8)
Mymensingh Division	313 (9.7)	127 (7.6)	76 (11.0)	85 (14.1)	17 (9.1)	8 (8.4)
Rajshahi Division	213 (6.6)	99 (5.9)	47 (6.8)	41 (6.8)	18 (9.6)	8 (8.4)
Rangpur Division	179 (5.5)	93 (5.6)	45 (6.5)	22 (3.7)	9 (4.8)	10 (10.5)
Sylhet Division	148 (4.6)	54 (3.2)	52 (7.6)	31 (5.1)	5 (2.7)	6 (6.3)
*Mother’s Level of Education*						
Higher education (above Bachelor)	1287 (39.7)	684 (41.0)	341 (49.5)	178 (29.5)	46 (24.6)	38 (40.)
	1514 (46.7)	961 (57.6)	226 (32.8)	233 (38.6)	67 (35.8)	27 (28.4)
Intermediate (11–12)	441 (13.6)	23 (1.4)	122 (17.7)	192 (31.84)	74 (39.5)	30 (31.6)
*Working status*						
Employed	2864 (88.3)	1647 (98.7)	600 (87.1)	432 (71.6)	99 (52.9)	86 (91.0)
Not employed/student	378 (11.7)	21 (1.3)	89 (12.9)	171 (28.4)	88 (47.1)	9 (9.47)
*Income in Taka*						
Lower-income (<30,000)	204 (6.3)	14 (0.9)	37 (5.4)	102 (16.9)	41 (21.9)	10 (10.4)
Middle-income (30,000–70,000)	1496 (46.1)	510 (30.9)	433 (62.8)	389 (64.5)	108 (57.8)	56 (59.0)
High-income (>70,000)	1542 (47.7)	1144 (68.59)	219 (31.8)	112 (18.8)	38 (20.3)	29 (30.5)
*Occupation*						
Healthcare workers	675 (20.8)	31 (1.9)	212 (30.8)	304 (50.4)	89 (47.6)	39 (41.1)
Non-healthcare worker	2567 (79.2)	1637 (98.1)	477 (69.2)	299 (49.6)	98 (52.4)	56 (59.0)

Prevalence and 95% confidence intervals of physical complaints by gender among internet users.

**Table 2 ijerph-18-10728-t002:** Gender differences in the prevalence of self-reported physical health scores among internet users in Bangladesh. Significant variables between males and females are bolded because the confidence intervals did not overlap.

Physical Complaints	*n*	Prevalence (95% CI)		
		Overall	Male	Female
**Back pain**	2261	69.9 [68.27, 71.43]	**78.14 [76.26, 79.91]**	56.90 [54.15, 59.62]
Finger numbness	2106	65.10 [63.42, 66.71]	**72.37 [70.35, 74.30]**	53.65 [50.89, 56.39]
Headaches	2346	72.50 [70.93, 74.01]	**78.80 [76.94, 80.54]**	62.62 [59.91, 65.25]
Inability to sleep	1396	43.14 [41.44, 44.85]	41.95 [39.79, 44.14]	45.00 [42.27, 47.76]
Poor Nutrition	746	23.05 [21.63, 24.54]	16.55 [14.97, 18.25]	**33.25 [30.70, 35.91]**
Poor Personal Hygiene	605	18.70 [17.39, 20.08]	14.98 [13.47, 16.62]	**24.52 [22.23, 26.98]**
Neck pain	1789	55.28 [53.56, 56.99]	**58.50 [56.31, 60.66]**	50.24 [47.48, 53.00]
Dry eyes/other vision problems	1815	56.09 [54.37, 57.79]	57.54 [55.35, 59.70]	53.81 [51.05, 56.55]
Weight gain/loss	1653	51.08 [49.36, 52.80]	52.43 [50.22, 54.63]	48.97 [46.21, 51.73]
Loss of Appetite	499	15.42 [14.22, 16.71]	13.77 [12.31, 15.36]	**18.02 [15.99, 20.24]**

CI = Confidence interval; *n*= number of respondents.

**Table 3 ijerph-18-10728-t003:** Mean of physical complaints depending on internet use. Results of separate simple linear regressions.

Variables	Mean Scores (SD)	Unadjusted Coefficient [95% CI]
**Gender**		
Male	4.85 (1.83) ***	Reference
Female	4.47 (1.88)	**−0.38 [−0.51, −0.25]**
**Age**		
18–27 yrs	3.52 (1.71) ***	Reference
28–37 yrs	5.02 (1.81)	**1.50 [1.33, 1.67]**
38+ yrs	4.80 (1.77)	**1.31 [1.12, 1.49]**
**Marital status**		
Single	3.67 (1.76) ***	Reference
Married	4.96 (1.79)	**1.30 [1.13, 1.47]**
Divorced/widow	3.00 (1.19)	**−0.67 [−1.04, −0.29]**
**Place of residence**		
Barisal Division	5.35 (1.83) ***	Reference
Chittagong Division	4.66 (1.87)	**−0.69 [−1.04, −0.35]**
Dhaka Division	4.36 (1.77)	**−0.99 [−1.28, −0.70]**
Khulna Division	4.83 (1.83)	**−0.52 [−0.85, −0.19]**
Mymensingh Division	5.03 (1.92)	−0.32 [−0.66, 0.02]
Rajshahi Division	5.21 (1.91)	−0.14 [−0.51, 0.22]
Rangpur Division	5.47 (1.88)	0.12 [−0.26, 0.50]
Sylhet Division	4.99 (1.83)	−0.36 [−0.76, 0.04]
**Mother’s Level of Education**		
Higher education	4.71 (1.82) ***	Reference
Bachelor	5.00 (1.87)	**0.30 [0.16, 0.43]**
Intermediate (11–12)	3.65 (1.50)	**−1.06 [−1.26, −0.86]**
**Working status**		
Employed	4.89 (1.79) ***	Reference
Not employed/student	3.24 (1.72)	**−1.65 [−1.84, −1.46]**
**Income in Taka**		
Lower-income (<30,000)	3.05 (1.50) ***	Reference
Middle-income (30,000)	4.82 (1.88)	**1.77 [1.51, 2.04]**
High-income (>70,000)	4.80 (1.77)	**1.75 [1.48, 2.01]**
**Occupation**		
Healthcare workers	3.62 (1.26) ***	Reference
Non-healthcare workers	4.98 (1.89)	**1.36 [1.21, 1.51]**

SD = standard deviation; CI = confidence interval. *** = *p* < 0.001 based on *t*-test and ANOVA. Bolded are significant coefficients.

**Table 4 ijerph-18-10728-t004:** Results of multiple regression analyses predicting the level of physical complaints among internet users under COVID-19 lockdown. Printed in bolds are significant predictors.

Variables	Adjusted Coefficients [95% CI]	*p*-Value
**Gender**		
Male	Reference	
Female	−0.08 [−0.20, 0.04]	0.179
**Age groups**		
18–27 yrs	Reference	
28–37 yrs	0.09 [−0.13, 0.30]	0.440
38+ yrs	−0.03 [−0.29, 0.23]	0.818
**Marital status**		
Single	Reference	
Married	−0.08 [−0.29, 0.12]	0.431
Divorced/widow	**−1.09 [−1.44, −0.74]**	<0.0005
**Place of residence**		
Barisal Division	Reference	
Chittagong Division	**−0.70 [−0.99, −0.41]**	<0.0005
Dhaka Division	**−1.27 [−1.51, −1.02]**	<0.0005
Khulna Division	**−0.42 [−0.70, −0.14]**	0.003
Mymensingh Division	**−0.31 [−0.59, −0.03]**	0.031
Rajshahi Division	−0.29 [−0.60, 0.01]	0.060
Rangpur Division	−0.11 [−0.43, 0.21]	0.487
Sylhet Division	**−0.40 [−0.74, −0.07]**	0.018
**Mother’s Level of Education**		
Higher education	Reference	
Bachelor	−0.00 [−0.16, 0.16]	0.976
Intermediate (11–12)	0.01 [−0.19, 0.21]	0.924
**Working status**		
Employed	Reference	
Not employed/student	**−0.49 [−0.72, −0.26]**	<0.0005
**Income in Taka**		
Lower-income (<30,000)	Reference	
Middle-income (30,000–70,000)	**0.72 [0.48, 0.96]**	<0.0005
High-income (>70,000)	0.26 [−0.00, 0.53]	0.052
**Occupation**		
Healthcare workers	Reference	
Non-health care worker	**0.62 [0.47, 0.78]**	<0.0005
**Internet use Frequency**		
Seldom	Reference	-
Casual	0.36 [−0.02, 0.74]	0.067
Regular	**0.52 [0.18, 0.85]**	0.003
Frequent	**1.21 [0.88, 1.54]**	<0.0005
Intense	**2.24 [1.91, 2.57]**	<0.0005
**Constant**	3.18 [2.69, 3.67]	<0.0005

The final model included Gender, Age group, Division, Mother’s education, Income status, Employment status, Occupational status, Marital status and Internet use frequency.

## Data Availability

Our data are included in the manuscript and raw data can be released at reasonable request.

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
