# Peer review of "Internet Use Impact on Physical Health during COVID-19 Lockdown in Bangladesh: A Web-Based Cross-Sectional Study"

_ijerph, 2021, doi:10.3390/ijerph182010728_

Round 1

Reviewer 1 Report

Internet Use Impact on Physical Health During COVID-19 2 Pandemic in Bangladesh: A Web-Based Cross-Sectional Study

Tanvir, Uchechukwu, Dewan ,(…) & Kingsley

Specific Comments to the Authors

This reviewer prefers making direct proposals over lengthy explanations

Line

Text / remarks / suggestions / proposals

Study time span

I take it that study participants have been surveyed in times of COVID-19 lockdown. In the text, authors should not alternate between “pandemic” and “lockdown”, because living in the pandemic was much less restricted than living under lockdown. It should be “lockdown” throughout the text, whenever the survey/ survey results are referred to.

Wording like “during the COVID-19” (spoken language) should not appear in a journal text, it should read “during COVID-19 lockdown”.

Generally

Spelling varies: COVID-19 Covid-19 / Internet internet / a variety of words are written with a capital letter but correctly in small letters in other parts of the text /

The questionnaire on physical complaints seems a bit arbitrary. F.i. why is there no question on digestive problems? Are the health problems addressed based on epidemiologic findings?

Tables

2,3, Suppl.1 --- letters under these tables are too small

Abstract

24 “physical health” clearer “self-reported physical health”

27 “Multiple linear regression…” clearer “Linear regression…”

29 A “multivariable” analysis is an unusual expression, better delete the word here, it is explained later in the Statistical analysis section

32 “Other important factors were gender…” Þ “Other important predictors were gender…” [factors imply some causality or at least have this connotation]

33 “Frequent and extensive use of the internet was strong predictors” Þ “were strong predictors”

90ff

“There may be more than one questionnaire filled by the same person that affects the statistical analysis results of questionnaire data.” Þ could be deleted, the information is given in 91ff

94

„…only can fill once.“ Þ  “…only could participate once.”

105

“… (Back and neck pain” Þ  “…[back and neck pain”

117

„Independent variables“ I would feel better if these were just called “demographic variables” (everywhere in the text)

124

As authors are not too precise in detailing their independent variables, they could at least add “See Table 1 for an overview and details.”

125ff

Statistical analyses (has a lot of unclear wording)

“(…) t-test used compares the means of two independent groups, One-way analysis of variance (one-way ANOVA) used  compares the means of more than two independent groups, correlation analyses, and regression analysis.” Þ  “(…) group means were compared by t-tests and one-way analysis of variance (ANOVA). Further, correlational analyses and regression analyses were conducted.”

“All confounding variables with a P > .20 were retained in the univariate linear regression analysis and were used to build a multivariable model [17]. A manual elimination procedure was applied for multivariate linear regression analyses to remove non-significant variables (P > .05).” Þ “In univariate linear regressions, all predictors with a P > .20 were retained and a multivariable model was build [see 17]. In multivariate linear regression analyses, an iterative manual elimination procedure was applied in order to remove non-significant variables (P > .05).” This is not a standard procedure. Why P >.20

“The main study factor variable (Internet usage frequency) was added to all significant confounding variables after elimination processes. The main study factor and independent variables associated with physical health scores (P < .05) were reported. In the regression analysis, we checked for homogeneity of variance and multicollinearity, including Variance Inflation Factors (VIF) and the VIF < 4 was considered suitable [18].” Þ “The main study variable of interest (internet usage frequency) was then included into the regression models in order to test for its predictive power.”

“In the regression analysis, we checked for homogeneity of variance and multicollinearity, including Variance Inflation Factors (VIF) and the VIF < 4 was considered suitable [18].” Þ  “For regression analyses, we checked for homogeneity of variance and multicollinearity, including variance inflation factors (VIF), variable samples with VIF values < 4 were considered suitable for regression analyses [18].”  --- I think authors did mean this?

But in sum, the data analyses are inadequate. Authors should conduct multivariate (simultaneous) regression analyses with model-building by statistical standard and give the usual information (goodness-of-fit indices, R2 variance explained). Authors’ mode of data analyses means accumulation of α-error, and there is no alpha-adjustment for multiple testing, and “manual eliminating procedure” is non-transparent, i.e. against the replicability requirement in research.

142 and passim

Authors sometimes write that a table would “depict” something  Þ  a little awkward a word, because depicting s.th. is to give a “flowery” description for something  Þ  “Table X gives the main demographic…” or “in Table X … the main demographic… is reported/ listed/ displayed”

145

„…participants were males (1260, 61.1%)…“  Þ  participants were males (1976, 61.1%)…“ according to Table 1

Table 1

„Higher education (above Bachelor „  Þ  „Higher education (above bachelor)”

171

“…health by intensive    internet…” [several superfluous blanks]  Þ  “…health by intensive internet…”

Table 2

Why are parts of the title in italics?

Figure 1

“Figure 1. Mean scores by gender of the physical health by internet use frequency in Bangladesh.”  Þ  “Figure 1. Mean scores of physical health under COVID-19 lockdown in Bangladesh, broken down by internet use and gender.” 

184

„those aged >28years“[blank is missing]   Þ   “those aged >28 years”

Table 3

“Table 3. Mean scores and Univariate linear regression of Physical Complaints among Internet Users.” Is not precise enough   Þ   “Table 3. Mean of physical complaints depending on internet use. Results of separate univariate linear regressions.”

199f

“…internet users during the COVID-19 had significantly higher physical health problems…”  Þ   “…internet users during the COVID-19 lockdown reported significantly higher physical health problems…” Because no medical diagnoses are involved in this survey.

211f

“assessing the relationship between problematic or addictive (excessive) use of the internet…”   Þ   “assessing the relationship between problematic, excessive, and addictive use of the internet…”  An internet use might be problematic, hazardous or excessive, but a pathological or addictive internet use would be very “heavy”. Still, it is not a medically recognized disorder in DSM-5 and ICD-10. (Internet gaming disorder is, but an internet use disorder is not). European readers are very much aware of that. So, I strongly advise that the term “internet addiction” should not be used in any part of the text.

232

“…which agrees with the findings of significant association…”  Þ   “…which is in line with the findings of significant association…”

262f

“…stayingoutside/working prior to the lockdown…”  Þ   “staying outside/working prior to the lockdown…” [blank missing]

292ff

Limitation paragraph. Authors could prompt other surveys, especially social representative studies using standardized quality-of-life measures, and/or studies addressing families (parents and children)…

304ff

“Despite these limitations, this was the first study to investigate the associations between extensive internet use and physical health symptoms in a sample of adolescents in mainland Bangladesh.” --- This sentence seems to be misplaced. This survey was not on adolescents, 18- to 27-years old are young adults (not adolescents).

319ff

“Arguably, it is time for the World Health Organization and health departments worldwide to develop effective health policies to increase public awareness…”   ---  There are activities like that proceeding in Germany (1). Therefore, it is better to say “it is time (…) to enhance effective…”  Authors’ wording imply quite an accusation. Are there really no WHO initiatives on this issue? See WHO website “Lower-income countries and indigenous populations receive WHO assistance amid the ongoing threat of COVID-19” and others.

(1)      Paschke, K., Holtmann, M., Melchers, P., Klein, M., Schimansky, G., Krömer, T., Reis, O., Wartberg, L., & Thomasius, R. (2020). Media-associated disorders in childhood and adolescence: Evidence paper of the joint addiction commission of the German societies and professional associations of child and adolescent psychiatry and psychotherapy. [Abstract in English]. Zeitschrift für Kinder- und Jugendpsychiatrie und Psychotherapie, 48(4):303-317. https://doi.org/10.1024/1422-4917/a000735

This reviewer is not involved in the above cited paper.

Table 1 (Suppl.)

“Table 1. Results of Multiple Regression Analysis in Prediction of Level of Physical Complaints 325 Among Internet Users. Bold are significant variables“  Þ  “Table 1. Results of multiple regression analyses predicting the level of physical complaints in internet users under COVID-19 lockdown. Printed in bolds are significant predictors“

References

Needs thorough reworking. Different referencing styles are mixed.

11 is a newspaper article, though published via internet.

Kalia, A. (2020). The Zoom Boom: How video calling became a blessing – and a curse. [Journalistic article ]. The Guardian 2020. URL: The Zoom boom: how video-calling became a blessing – and a curse | Zoom | The Guardian

18 Vatcheva, K.P.; Lee, M.; McCormick, J.B.; Rahbar, M.H. Multicollinearity in regression analyses conducted in epidemiologic studies. Epidemiology (Sunnyvale 2016, 6, 41722161–1165 1000227.

(Sunnyvale) 2016, 6, 41722161–1165 1000227. --- bracket was missing

31 Husain --- published online, URL is missing.

34, 35 --- citations incomplete.

Author Response

Reviewer #1:

Comments

Response

Study time span

I take it that study participants have been surveyed in times of COVID-19 lockdown. In the text, authors should not alternate between “pandemic” and “lockdown”, because living in the pandemic was much less restricted than living under lockdown. It should be “lockdown” throughout the text, whenever the survey/ survey results are referred to.

All references to pandemics have been replaced with lockdown across the manuscript

Wording like “during the COVID-19” (spoken language) should not appear in a journal text, it should read “during COVID-19 lockdown”.

‘During COVID-19’ have been used across the manuscript

Generally

Spelling varies: COVID-19 Covid-19 / Internet internet / a variety of words are written with a capital letter but correctly in small letters in other parts of the text /

Done. Internet was used.

The questionnaire on physical complaints seems a bit arbitrary. F.i. why is there no question on digestive problems? Are the health problems addressed based on epidemiologic findings?

The conditions reported in the study was based on those reported in other study (Zheng, et al. 2016) but from medical viewpoint, digestive conditions such as irritable bowel syndrome and Crohn's disease could cause a loss of appetite.

Table

Tables2,3, Suppl.1 --- letters under these tables are too small

Done. Letters under the table made larger than before.

Abstract

24 “physical health” clearer “self-reported physical health”

Change was made in this section and other relevant sections.

27 “Multiple linear regression…” clearer “Linear regression….”

Agreed and changes have been made throughout the manuscript

29 A “multivariable” analysis is an unusual expression, better delete the word here, it is explained later in the Statistical analysis section

Agreed and now reported as Multiple linear regression

32 “Other important factors were gender…” Þ “Other important predictors were gender…” [factors imply some causality or at least have this connotation]

Done. Sentence revised.

33 “Frequent and extensive use of the internet was strong predictors” Þ “were strong predictors”

Done. Revised

90ff

“There may be more than one questionnaire filled by the same person that affects the statistical analysis results of questionnaire data.” Þ could be deleted, the information is given in 91ff

Done. The referred section was deleted

94

„…only can fill once.“ Þ  “…only could participate once.”

Done. Revised

105

“… (Back and neck pain” Þ  “…[back and neck pain”

Done. Revised

117

„Independent variables“ I would feel better if these were just called “demographic variables” (everywhere in the text)

Demographic variables have been used across the manuscript.

124

As authors are not too precise in detailing their independent variables, they could at least add “See Table 1 for an overview and details.”

Done. We have referenced Table 1 as suggested.

125ff

Statistical analyses (has a lot of unclear wording)

“(…) t-test used compares the means of two independent groups, One-way analysis of variance (one-way ANOVA) used  compares the means of more than two independent groups, correlation analyses, and regression analysis.” Þ  “(…) group means were compared by t-tests and one-way analysis of variance (ANOVA). Further, correlational analyses and regression analyses were conducted.”

Response: Thanks for the feedback and we have adjusted as suggested to read better, but we did not perform correlational analyses.

“All confounding variables with a P > .20 were retained in the univariate linear regression analysis and were used to build a multivariable model [17]. A manual elimination procedure was applied for multivariate linear regression analyses to remove non-significant variables (P > .05).” Þ “In univariate linear regressions, all predictors with a P > .20 were retained and a multivariable model was build [see 17]. In multivariate linear regression analyses, an iterative manual elimination procedure was applied in order to remove non-significant variables (P > .05).” This is not a standard procedure. Why P >.20

Response: P < 0.20 is a prescreening method before carrying out a multiple regression analysis. This prescreening approach has been used and reported in this journal and others (Ghimire et al., 2020 and Ezeh et al., 2021).  In some published articles, a less stringent threshold, such as P-value <0.25, has been used (Grant et al. 2019). We did not do an iterative (repetition) approach; instead, we did a manual elimination procedure to do a Wald test to determine non-significant variables.

References:

Grant, S. W., Hickey, G. L., & Head, S. J. (2019). Statistical primer: multivariable regression considerations and pitfalls. European Journal of Cardio-Thoracic Surgery, 55(2), 179-185.

Ghimire, P. R., Akombi-Inyang, B. J., Tannous, C., & Agho, K. E. (2020). Association between obesity and miscarriage among women of reproductive age in Nepal. Plos one, 15(8), e0236435.

Ezeh, O.K.; Ogbo, F.A.; Odumegwu, A.O.; Oforkansi, G.H.; Abada, U.D.; Goson, P.C.; Ishaya, T.; Agho, K.E. Under-5 Mortality and Its Associated Factors in Northern Nigeria: Evidence from 22,455 Singleton Live Births (2013–2018). Int. J. Environ. Res. Public Health 2021, 18, 9899. https://doi.org/10.3390/ijerph18189899

“The main study factor variable (Internet usage frequency) was added to all significant confounding variables after elimination processes. The main study factor and independent variables associated with physical health scores (P < .05) were reported. In the regression analysis, we checked for homogeneity of variance and multicollinearity, including Variance Inflation Factors (VIF) and the VIF < 4 was considered suitable [18].” Þ “The main study variable of interest (internet usage frequency) was then included into the regression models in order to test for its predictive power.”

Done. We included “The main study variable of interest (internet usage frequency)” to examine Impact or association. 

“In the regression analysis, we checked for homogeneity of variance and multicollinearity, including Variance Inflation Factors (VIF) and the VIF < 4 was considered suitable [18].” Þ  “For regression analyses, we checked for homogeneity of variance and multicollinearity, including variance inflation factors (VIF), variable samples with VIF values < 4 were considered suitable for regression analyses [18].”  --- I think authors did mean this?

Done. The suggested corrections have been implemented.

But in sum, the data analyses are inadequate. Authors should conduct multivariate (simultaneous) regression analyses with model-building by statistical standard and give the usual information (goodness-of-fit indices, R2 variance explained). Authors’ mode of data analyses means accumulation of α-error, and there is no alpha-adjustment for multiple testing, and “manual eliminating procedure” is non-transparent, i.e. against the replicability requirement in research.

We understand that R-square increases when extra explanatory variables are added to the model (changes in the dependent variable) because it explains how well the model fits the data. Still, we don’t think that is the aim of this research. However, Impact measures are especially common in epidemiology and frequently quantify relationships between exposures and outcomes.

142 and passim

Authors sometimes write that a table would “depict” something  Þ  a little awkward a word, because depicting s.th. is to give a “flowery” description for something  Þ  “Table X gives the main demographic…” or “in Table X … the main demographic… is reported/ listed/ displayed”

Done. sentence was revised

145

„…participants were males (1260, 61.1%)…“  Þ  participants were males (1976, 61.1%)…“ according to Table 1

Done. sentence was revised

Table 1

„Higher education (above Bachelor „  Þ  „Higher education (above bachelor)”

Done. sentence was revised

171

“…health by intensive    internet…” [several superfluous blanks]  Þ  “…health by intensive internet…”

Done. sentence was revised

Table 2

Why are parts of the title in italics?

Done. sentence was revised

Figure 1

“Figure 1. Mean scores by gender of the physical health by internet use frequency in Bangladesh.”  Þ  “Figure 1. Mean scores of physical health under COVID-19 lockdown in Bangladesh, broken down by internet use and gender.” 

Done. sentence was revised

184

„those aged >28years“[blank is missing]   Þ   “those aged >28 years”

Done. sentence was revised

Table 3

“Table 3. Mean scores and Univariate linear regression of Physical Complaints among Internet Users.” Is not precise enough   Þ   “Table 3. Mean of physical complaints depending on internet use. Results of separate univariate linear regressions.”

Done. sentence was revised

199f

“…internet users during the COVID-19 had significantly higher physical health problems…”  Þ   “…internet users during the COVID-19 lockdown reported significantly higher physical health problems…” Because no medical diagnoses are involved in this survey.

Done. sentence was revised

211f

“assessing the relationship between problematic or addictive (excessive) use of the internet…”   Þ   “assessing the relationship between problematic, excessive, and addictive use of the internet…” 

Done. sentence was revised

An internet use might be problematic, hazardous or excessive, but a pathological or addictive internet use would be very “heavy”. Still, it is not a medically recognized disorder in DSM-5 and ICD-10. (Internet gaming disorder is, but an internet use disorder is not). European readers are very much aware of that. So, I strongly advise that the term “internet addiction” should not be used in any part of the text.

Done. The term ‘internet addiction has been revised to excessive use

232

“…which agrees with the findings of significant association…”  Þ   “…which is in line with the findings of significant association…”

Done. sentence was revised

262f

“…stayingoutside/working prior to the lockdown…”  Þ   “staying outside/working prior to the lockdown…” [blank missing]

Done. sentence was revised

292ff

Limitation paragraph. Authors could prompt other surveys, especially social representative studies using standardized quality-of-life measures, and/or studies addressing families (parents and children)…

Agreed, but the questions used in this study was from a previous study (Zheng et al. 2016)

Zheng, Y., Wei, D., Li, J., Zhu, T., & Ning, H. (2016). Internet use and its impact on individual physical health. IEEE Access4, 5135-5142.

304ff

“Despite these limitations, this was the first study to investigate the associations between extensive internet use and physical health symptoms in a sample of adolescents in mainland Bangladesh.” --- This sentence seems to be misplaced. This survey was not on adolescents, 18- to 27-years old are young adults (not adolescents).

Done. Adults have been used.

“Arguably, it is time for the World Health Organization and health departments worldwide to develop effective health policies to increase public awareness…”   ---  There are activities like that proceeding in Germany (1). Therefore, it is better to say “it is time (…) to enhance effective…” 

Authors’ wording imply quite an accusation. Are there really no WHO initiatives on this issue? See WHO website “Lower-income countries and indigenous populations receive WHO assistance amid the ongoing threat of COVID-19” and others.

Done. Revised

319ff

(1)      Paschke, K., Holtmann, M., Melchers, P., Klein, M., Schimansky, G., Krömer, T., Reis, O., Wartberg, L., & Thomasius, R. (2020). Media-associated disorders in childhood and adolescence: Evidence paper of the joint addiction commission of the German societies and professional associations of child and adolescent psychiatry and psychotherapy. [Abstract in English]. Zeitschrift für Kinder- und Jugendpsychiatrie und Psychotherapie, 48(4):303-317. https://doi.org/10.1024/1422-4917/a000735

This reviewer is not involved in the above cited paper.

We appreciate the sharing of this paper with us and have revised our manuscript in line with this.

Table 1 (Suppl.)

“Table 1. Results of Multiple Regression Analysis in Prediction of Level of Physical Complaints 325 Among Internet Users. Bold are significant variables“  Þ  “Table 1. Results of multiple regression analyses predicting the level of physical complaints in internet users under COVID-19 lockdown. Printed in bolds are significant predictors“

Done in Supplementary table 1

References

Needs thorough reworking. Different referencing styles are mixed.

The correct referencing style has been used.

11 is a newspaper article, though published via internet.

Kalia, A. (2020). The Zoom Boom: How video calling became a blessing – and a curse. [Journalistic article ]. The Guardian 2020. URL: The Zoom boom: how video-calling became a blessing – and a curse | Zoom | The Guardian

Reference has been accurately edited

18 Vatcheva, K.P.; Lee, M.; McCormick, J.B.; Rahbar, M.H. Multicollinearity in regression analyses conducted in epidemiologic studies. Epidemiology (Sunnyvale 2016, 6, 41722161–1165 1000227.

(Sunnyvale) 2016, 6, 41722161–1165 1000227. --- bracket was missing

Done. Revised

31 Husain --- published online, URL is missing.

Changed the reference. 31 Husain has been removed and replaced

32.       Abir, T.; Kalimullah, N.A.; Osuagwu, U.L.; Nur-A Yazdani, D.M.; Husain, T.; Goson, P.C.; Basak, P.; Rahman, M.A.; Al Mamun, A.; Permarupan, P.Y.; et al. Prevalence and factors associated with mental health impact of COVID-19 pandemic in Bangladesh: A survey-based cross-sectional study. Ann. Glob. Health, 2021, 87, 43.

More accurate and specific references used.

34, 35 --- citations incomplete.

Done. Revised.

Reviewer 2 Report

The authors propose the use of a multiple linear regression model to understand the impact on physical health during COVID-19 pandemic in Bangladesh. A questionnaire was applied online. While not ideal, it was a good idea to block IP addresses trying to avoid multiple answers from the same person. A huge drawback (as stated by the authors) is since all data were collected directly online then people from remote regions might not have the opportunity to fill such questionnaire.

Although this is a very interesting topic and may have a high interest between the readers, I do have some concerns regarding the applied methodology and, consequently, the inferences.

Major comments

  1. The response variable is based on 10 items (yes/no response) from the questionnaire regarding individual physical health on internet use. As stated by the authors "Binary scores for individual questions were summed to give a physical health score which ranged from 0-10". Authors then applied the multiple linear regression model in order to understand the behaviour of such sum based on some explanatory variables. My major concern with this manuscript is that multiple linear regression may not be the most appropriate model to describe such response, since this sum is a discrete value and then there is a violation in the basic assumptions of such model. For instance, one better possible alternative would be to categorise this sum in three different groups and then fit a multinomial regression model.
  2. Do the authors have the full information regarding the age of the respondents or the variable age was indeed collected as a factor? If the full information is available, authors must use it to avoid possible misinformation and lose of available (and important) data. 

  3. If multiple linear regression is in use, authors must provide a richer exploratory variable, presenting box plots from each explanatory variable against the response.

  4. Regardless of the considered model, authors must provide a full residual analysis in order to check all model assumptions and hence, to have robust and reliable inferences.

Minor comments

  1. Authors may replace the term 'multivariable model' and 'multivariate linear regression' for 'multiple linear regression' throughout the text. Multivariate stands for more than one response variable, which is not the case.
  2. Authors may replace the term 'univariate linear regression' for 'simple linear regression'.

Author Response

Reviewer #2:

Comments

Responses

1

The authors propose the use of a multiple linear regression model to understand the impact on physical health during COVID-19 pandemic in Bangladesh. A questionnaire was applied online. While not ideal, it was a good idea to block IP addresses trying to avoid multiple answers from the same person. A huge drawback (as stated by the authors) is since all data were collected directly online then people from remote regions might not have the opportunity to fill such questionnaire.

Although this is a very interesting topic and may have a high interest between the readers, I do have some concerns regarding the applied methodology and, consequently, the inferences.

Thanks for the comments. We have provided a point-by-point response to the comments below

Major Comment

1

The response variable is based on 10 items (yes/no response) from the questionnaire regarding individual physical health on internet use. As stated by the authors "Binary scores for individual questions were summed to give a physical health score which ranged from 0-10". Authors then applied the multiple linear regression model in order to understand the behaviour of such sum based on some explanatory variables. My major concern with this manuscript is that multiple linear regression may not be the most appropriate model to describe such response, since this sum is a discrete value and then there is a violation in the basic assumptions of such model. For instance, one better possible alternative would be to categorize this sum in three different groups and then fit a multinomial regression model.

Thanks for the suggestion. There is no best way to go about the regression model for 10 point scores without some drawbacks. Using three different groups and then fit a multinomial regression model also have some limitations. (a) information is lost, and hence statistical power to detect an association is reduced, including underestimating the extent of variation (Altman & Royston, 2006) and (b)there is no intrinsic ordering when using multinomial logistic regression. In this study, we employed a statistical method similar to those carried out by Zhang et al. 2016. However, we have included the text below as part of the limitation of this study.

This study is also limited because using a 0-10 points physical health score may violate some linear regression assumptions (Knief, 2020).

REF:

Altman, D. G., & Royston, P. (2006). The cost of dichotomizing continuous variables. Bmj, 332(7549), 1080.

Zheng, Y., Wei, D., Li, J., Zhu, T., & Ning, H. (2016). Internet use and its impact on individual physical health. IEEE Access, 4, 5135-5142.

Knief, U., & Forstmeier, W. (2021). Violating the normality assumption may be the lesser of two evils. Behavior Research Methods, 1-15.

2

Do the authors have the full information regarding the age of the respondents or the variable age was indeed collected as a factor? If the full information is available, authors must use it to avoid possible misinformation and lose of available (and important) data. 

We have the age for all participants and have provided the mean age in the manuscript (page 5, line 157). Categorization of Age is for public health purposes, as when patients are classified into, for example, pediatric, adult, and geriatric groups for healthcare resource allocation and for the everyday descriptive purposes used in the allied context, for as when people are referred to as being in their early twenties or mid-forties. No changes were made. For example in https://www.ncbi.nlm.nih.gov/pmc/articles/PMC4746832/; https://www.ncbi.nlm.nih.gov/pmc/articles/PMC5806344/

We have also included this explanation in the manuscript

3

If multiple linear regression is in use, authors must provide a richer exploratory variable, presenting box plots from each explanatory variable against the response.

Good point but not necessary for this paper because it will change the manuscript's focus, which is on “Internet Use Impact on Physical Health.”

4

Regardless of the considered model, authors must provide a full residual analysis in order to check all model assumptions and hence, to have robust and reliable inferences.

We have provided further analysis, and our insignificant result from the Breusch-Pagan test indicates a lack of heteroskedasticity- meaning the presence of equal variance of the residuals along the predicted line (homoskedasticity). In addition, we conducted some preliminary analysis using summary statistics, and we found the mean and median are almost similar, and the skewness was close to zero (Knief & Forstmeier, 2021).

Breusch-Pagan / Cook-Weisberg test for heteroskedasticity

Ho: Constant variance

Variables: fitted values of psych

chi2(1)      =     2.77

Prob > chi2  =   0.0960

As indicated above, we have included the text below in the limitation section of this paper

This study is also limited because using a 0-10 points physical health score may violate some linear regression assumptions (Knief, 2020).

Knief, U., & Forstmeier, W. (2021). Violating the normality assumption may be the lesser of two evils. Behaviour Research Methods, 1-15.

Minor Comment

1

Throughout the text, authors may replace the term 'multivariable model' and 'multivariate linear regression' for 'multiple linear regression'. Multivariate stands for more than one response variable, which is not the case.

Done. Line 140

2

Authors may replace the term 'univariate linear regression' for 'simple linear regression'.

Done across the manuscript. E.g., see Table 3 heading

Reviewer 3 Report

- Sampling technique is required along with the actual size of sample.

- this paper is more of qualitative research realm; so in results, discussion the data collected from focused group interview needs to be given more emphasis than document analysis.

- the real life situations and challenges of the sample need to be highlighted more; this would be a better ground to give suggestive measures in conclusion part.

- the paper is nicely drafted.

Author Response

Reviewer #3:

Comments

responses

1

Sampling technique is required along with the actual size of sample.

Total sample was 3242

We indicated “convenience sampling” in the manuscript – see line 87

2

- this paper is more of qualitative research realm; so in results, discussion the data collected from focused group interview needs to be given more emphasis than document analysis.

Thank you for your comment, and we have included qualitative research as future research.

3

- the real life situations and challenges of the sample need to be highlighted more; this would be a better ground to give suggestive measures in conclusion part.

We have included those as part of the five limitations of the study

Round 2

Reviewer 1 Report

ijerph-1370546-peer-review-v2

Internet Use Impact on Physical Health During COVID-19 Lockdown in Bangladesh: A Web-Based Cross-Sectional Study

Tanvir, Uchechukwu, Dewan , (…) & Kingsley

2nd review

Specific Comments

This reviewer prefers making direct proposals over lengthy explanations

Line

Text / remarks / suggestions / proposals

Generally

My remarks [and the other reviewers’ remarks] on V1.0 have been considered which puts the article on a proper IJERPH level I thank the authors for recommending literature tosome of my questions.

Spaces

Throughout the text, large spaces occur between words, e.g. lines 86, 88, 96 and a lot more; at the right end of lines and easy to spot; mostly in the context of hyphens or separators

117 (See   => (see                         on various places in the text

§ Demographic variables  => all words in [squared brackets] should be written in small letters here

128 One-way analysis of variance  => one-way analysis of variance

Table 1 (title) No word but ‘Bangladesh’ should be written with a capital letter here

Higher education (above bachelor => (above bachelor) close bracket after ‘bachelor’ missing

Page 7, within Table 2: spelling and brackets

lower-income (<30,000 =>          Lower-income (<30,000)

middle-income (30,000 =>          Middle-income (30,000)

There is a P in the note line that is too big in print

line 210 “Demographic variables adjusted for are Gender, Age group, Division of residence, Mothers Education, marital status, income, working status, occupation.”

Demographic variables adjusted for are gender, age group, division of residence, mother’s education, marital status, income, working status, occupation.

Reviewer 2 Report

Authors have addressed all my points. 
